# The improvement of the shear stress and oscillatory shear index of coronary arteries during Enhanced External Counterpulsation in patients with coronary heart disease

Ling Xu[1], Xi Chen[2], Ming Cui[1], Chuan Ren[1], Haiyi Yu[1], Wei Gao[1], Dongguo Li[2]*, Wei Zhao[1]*

1 NHC Key Laboratory of Cardiovascular Molecular Biology and Regulatory Peptides, Department of Cardiology, Peking University Third Hospital, Beijing, China, 2 School of Biomedical Engineering, Capital Medical University, Beijing, China

☯ These authors contributed equally to this work.
* ldg213@ccmu.edu.cn (DGL); beate_vv@bjmu.edu.cn (WZ)

**Data Availability Statement:** All relevant data are within the manuscript and Supporting Information files.

## Abstract

### Background

Enhanced External Counterpulsation (EECP) can chronically relieve ischemic chest pain and improve the prognosis of coronary heart disease (CHD). Despite its role in mitigating heart complications, EECP and the mechanisms behind its therapeutic nature, such as its effects on blood flow hemodynamics, are still not fully understood. This study aims to elucidate the effect of EECP on significant hemodynamic parameters in the coronary arterial tree.

### Methods

A finite volume method was used in conjunction with the inlet pressure wave (surrogated by the measured aortic pressure) before and during EECP and outlet flow resistance, assuming the blood as newtonian fluid. The time-average wall shear stress (TAWSS) and oscillatory shear index (OSI) were determined from the flow field.

### Results

Regardless of the degree of vascular stenosis, hemodynamic conditions and flow patterns could be improved during EECP. In comparison with the original tree, the tree with a severe stenosis (75% area stenosis) demonstrated more significant improvement in hemodynamic conditions and flow patterns during EECP, with surface area ratio of TAWSS risk area (SAR-TAWSS) reduced from 12.3% to 6.7% (vs. SAR-TAWSS reduced from 7.2% to 5.6% in the original tree) and surface area ratio of OSI risk area (SAR-OSI) reduced from 6.8% to 2.5% (vs. SAR-OSI of both 0% before and during EECP in the original tree because of mild stenosis). Moreover, it was also shown that small ratio of diastolic pressure (D) and systolic

**Funding:** This Paper has been partly funded by National Natural Science Foundation of China (Grant No. 81601968, Wei Zhao) and National Natural Science Foundation of China (Grant No. 11802187, Xi Chen).

**Competing interests:** The authors have declared that no competing interests exist.

**Abbreviations:** EECP, Enhanced External Counterpulsation; CHD, Coronary Heart Disease; LMCA, Left Main Coronary Arterial; TAWSS, Time-Average Wall Shear Stress; OSI, Oscillatory Shear Index; SAR-TAWSS, Oscillatory Shear Index; D/S, the ratio of diastolic pressure (D) and systolic pressure (S); PCI, Percutaneous Coronary Intervention; CFD, Computational Fluid Dynamics; FFR, Fractional Flow Reserve; CTA, Computer Tomography Angiography; LAD, Left Anterior Descending Artery.

pressure (S) (D/S) could only improve the hemodynamic condition mildly. The SAR-TAWSS reduction ratio significantly increased as D/S became larger.

## Conclusions

A key finding of the study was that the improvement of hemodynamic conditions along the LMCA trees during EECP became more significant with the increase of D/S and the severity degree of stenoses at the bifurcation site. These findings have important implications on EECP as adjuvant therapy before or after percutaneous coronary intervention (PCI) in patients with diffuse atherosclerosis.

## Introduction

Enhanced External Counterpulsation (EECP) can noninvasively assist circulation in a safe and effective way. EECP increases blood pressure during diastole and causes reversal of blood flow direction in systole, thus generating a unique shape of aortic pressure wave [1]. In addition to the immediate effects of EECP, some patients also experience sustained benefits which can last for up to 5 years post-therapy. Therefore, some persistent mechanisms underlying it could exist [2]. According to latest researches, the increase of shear stress may explain this phenomenon [3–7].

In 2007, Zhang et al discovered that the shear stress in the EECP group was significantly higher than the baseline and the control group in a model of hypercholesterolemic pigs [3]. Later, Du et al used 3-D fluid structure interaction technology to rebuild the vasculature and monitored the shear stress of hypercholesterolemic pig in vitro, finding that both the plague wall stress and the time average wall shear stress significantly increased after EECP treatment [4]. Recently, in healthy volunteers, Randy et al showed that shear stress in both brachial and femoral arteries increased during EECP [5]. Based on the above, it is assumed that EECP can promote long-time relief from ischemic chest pain and improve the prognosis of coronary heart disease (CHD) by increasing shear stress. Additionally, the ratio of diastolic pressure (D) and systolic pressure (S) (D/S) was an important parameter of EECP, and it directly determined the increase of blood pressure during diastole [6,7]. However, previous work did not present the effects of the important parameters on hemodynamic improvement during EECP (i.e., the severe degree of stenosis and the effect of D/S).

Time-averaged wall shear stress (TAWSS) and oscillatory shear index (OSI) are well known as primary risk parameters for the development and progression of atherosclerosis [8–12], which can further lead to various types of coronary stenosis [13]. Recent studies have discovered that low TAWSS ($\leq$ 4 dynes/cm$^2$) and high OSI ($\geq$ 0.15) are risk factors for rupture-prone phenotype, which may be related to lipid accumulation and inflammatory cell infiltration to the intima [14–19]. Therefore, the evaluation of hemodynamic parameters in the epicardial coronary arterial tree is very important for understanding the progression of atherosclerosis as well as high-risk plaque formation. To evaluate the efficacy of EECP as adjuvant therapy after stent implantation or coronary artery bypass grafting, certain hemodynamic parameters can be applied.

Computational fluid dynamics (CFD) methods have been widely used in conjunction with empirically measured waveforms (as boundary conditions) to predict blood flow disturbances (e.g., flow separation, secondary flow, stagnation point flow, reversed flow, and/or turbulence) caused by convective inertia [12, 20–22], using TAWSS and OSI as important parameters [11,

19–21]. Recently, the CFD methods have been adopted to non-invasively determine the fractional flow reserve (FFR) [23, 24], which may guide percutaneous coronary intervention (PCI) for a better clinical outcome [25, 26]. However, few researchers have investigated the effect of EECP based on the CFD methods through examining hemodynamic changes [27].

The objective of this study is to investigate the hemodynamic changes in the patient-specific epicardial left main coronary arterial (LMCA) tree before or during EECP. Hemodynamic parameters used included TAWSS and OSI (SAR-TAWSS and SAR-OSI), and the flow fields were presented. In addition, the effects of other important parameters were also investigated, such as the severity degree of stenosis and D/S. Finally, the significance and limitations of these simulations were discussed.

## Materials and methods

### Study design

Seven human subjects (six with stent implantation and one with coronary artery bypass grafting) underwent computer tomography angiography (CTA) of coronary arteries. Morphometric data of the epicardial LMCA tree was reconstructed from CTA images. The three-dimensional geometrical model was meshed, and the Navier-Stokes and continuity equations were solved using a transient finite volume method. The inlet boundary conditions were the aortic pressure waves before and during EECP. The outlet boundary conditions were flow resistances.

### Ethics statement

This is an observational, retrospective study which was performed in compliance with the principles outlined in the Declaration of Helsinki and approved by the Ethics Committee of Peking University Third Hospital and all patients had signed informed consent.

### Imaging acquisition

All studies were performed on a dual-source CT scanner (Siemens Definition, Forchheim, Germany). After an initial survey scan, a retrospectively gated contrast-enhanced scan was performed using 80 ml of iodinated contrast (Iopromide-Ultravist 370, Bayer Healthcare, Morristown, USA) injection through an antecubital vein at 5 ml/s followed by 50 ml of normal saline at the same rate. The scan parameters were: $2 \times 64 \times 0.6$ mm collimation, tube voltage– 120 kV; tube current–average 620 mAs adjusted to body size; gantry rotation time– 330 msec; pitch– 0.2–0.43 depending on heart rate. The simultaneous acquisition of multi-parallel cross sections enabled the imaging of coronary artery in a single breath hold. Images were reconstructed with a slice thickness/increment of 0.7/0.4 mm with B26f at temporal resolution of 83 msec (half-scan). The initial data window was positioned at 70% of the R-R interval, with additional data sets reconstructed at ±5% intervals to compensate for motion artifacts in coronary arteries if necessary.

In order to get the aortic pressure, pulse wave analysis (SphygmoCor Version 9, AtCor Medical Pty. Ltd, Australia) was performed on each patient before and during EECP treatment. Pressure oscillations generated by brachial arterial pulsation are transmitted to brachial blood pressure cuff, measured by a transducer and then fed into a microprocessor. Computerized software records pulse wave of brachial artery and derives central aortic pulse wave with a validated generalized transfer factor.

## Geometrical models

Morphometric data of epicardial LMCA trees were extracted from patients' CTA images using the MIMICS software (Materialise, NV, Belgium). Based on the morphometric data, geometrical models were generated using the Geomagic Studio software (3D Systems, Rock Hill, USA) and then meshed using ANSYS ICEM (ANSYS Inc., Canonsburg, USA), as shown in Fig 1A and 1B. A mesh dependency was conducted such that the relative error in two consecutive mesh refinements was < 1% for the maximum velocity of steady state flow with inlet flow velocity equal to the time-averaged velocity over a cardiac cycle. A total of approximately 500,000 tetrahedral shaped volume elements (element size = 0.2 mm) were necessary to accurately mesh the computational domain.

## 3-D computational model

The governing equations were formulated for coronary arteries, each vessel of which was assumed to be rigid and impermeable. Navier-Stokes and continuity equations were solved using the commercial software solver FLUENT (ANSYS, Inc., Canonsburg, USA). Similar to previous studies [28], three cardiac cycles were required to achieve convergence for the transient analysis. The implicit Euler method was used and a constant time step was employed,

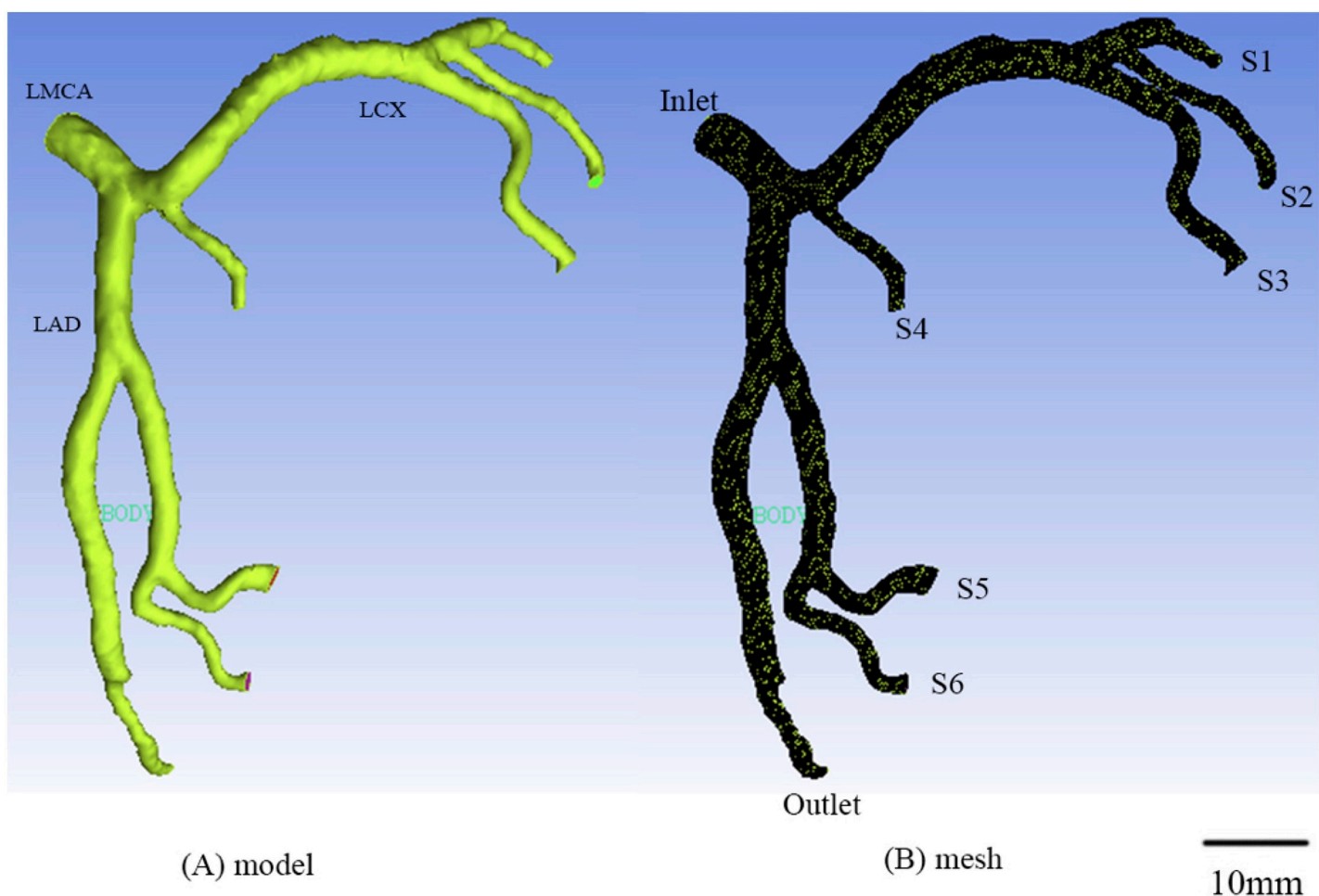

**Fig 1.** Geometrical model reconstructed from CTA (A) and computational meshes (B) in the epicardial LMCA tree of a representative patient.

where Δt = 0.01 s with 80 total time step per cardiac cycle. Although blood is a suspension of particles, it behaves as a Newtonian fluid in vessels with diameters > 1 mm [29]. The measured aortic pressure waves before or during EECP (Fig 2A and 2E) were set as the boundary condition at the inlet of LMCA trees. It was assumed that the distribution of the resting blood flow in normal coronary arteries obeyed scale laws [30], so the resistance of each coronary outlet ($R_i$) was computed by the total coronary resistance ($R_{inlet}$) and a morphometry factor ($N_i$), which was inversely related to the branch diameters [30, 31]. Many physiological studies have shown that the coronary pressure-flow lines were concave toward the flow axis at lower pressures [32,33], and the zero flow pressure intercept at the physiological pressure range (i.e., $P_0$ in this study) exceeded coronary venous or left ventricular diastolic pressure by five to ten-fold [34]. Therefore, $P_0$ was chosen to be 51.7 mmHg [34] to determine $P_i$ (the pressure at each outlet) and $R_i$ through iterative procedures (see details in S1 Appendix), which was similar to a previous study [35]. The viscosity (μ) and density (ρ) of the solution were assumed as $4.5 \times 10^{-3}$ Pa·s and 1060 kg/m³ respectively to mimic blood flow with a hematocrit of about 45% in these arteries [36]. After the velocity and pressure of the blood flow were calculated, hemodynamic parameters including TAWSS and OSI were determined from the equations in the S1

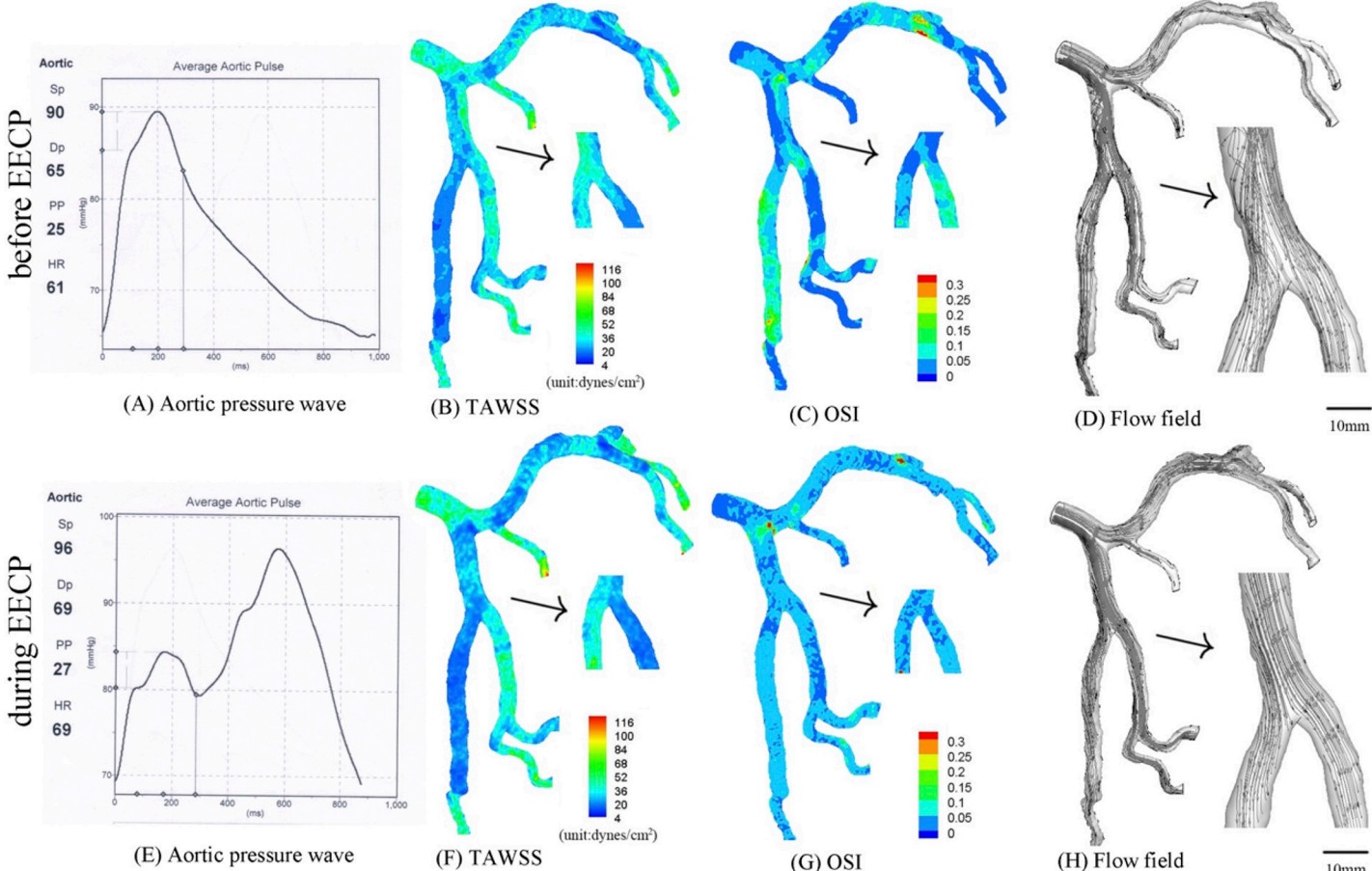

**Fig 2.** (A-D) Measured aortic pressure wave (A), TAWSS (B), OSI (C) and flow field (D) in the epicardial LMCA tree of a representative patient before EECP; (E-H) measured aortic pressure wave (E), TAWSS (F), OSI (G) and flow field (H) in the epicardial LMCA tree of the patient during EECP. The small figures for TAWSS and OSI show the posterior view. The small figures for flow field show the zoomed view.

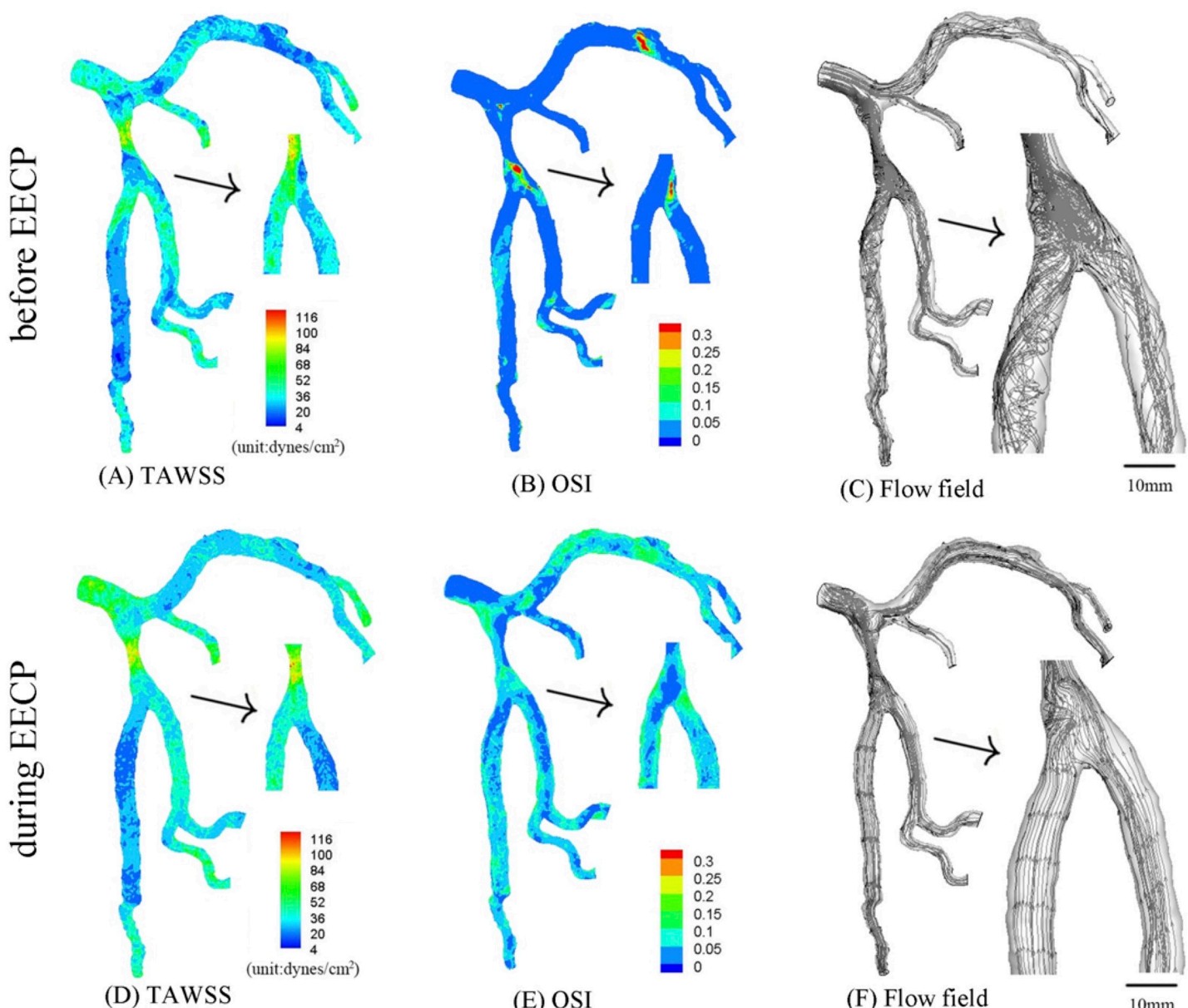

**Fig 3.** In correspondence with Fig 1A, TAWSS (A), OSI (B) and flow field (C) in the epicardial tree that has an idealized 75% area stenosis at the parent vessel (stenotic length of 7.3 mm) in the first bifurcation of LAD arterial tree before EECP; TAWSS (D), OSI (E) and flow field (F) in the same tree during EECP. The small figures for TAWSS and OSI show the posterior view. The small figures for flow field show the zoomed view.

Appendix. Moreover, SAR-TAWSS (surface area ratio of TAWSS that equals to

$$\frac{\text{Surface area}_{\text{TAWSS} \leq 4 \text{ dynes} \cdot \text{cm}^{-2}}}{\text{Surface area near a bifurcation}} \times 100\% \tag{1}$$

, where surface area near a bifurcation denotes 10mm length from the distal bifurcation to daughter vessels, and surface area of TAWSS $\leq$ 4 dynes/cm$^2$ is high-risk area which may induce coronary heart disease [14–19]) and SAR-OSI (surface area ratio of high OSI that

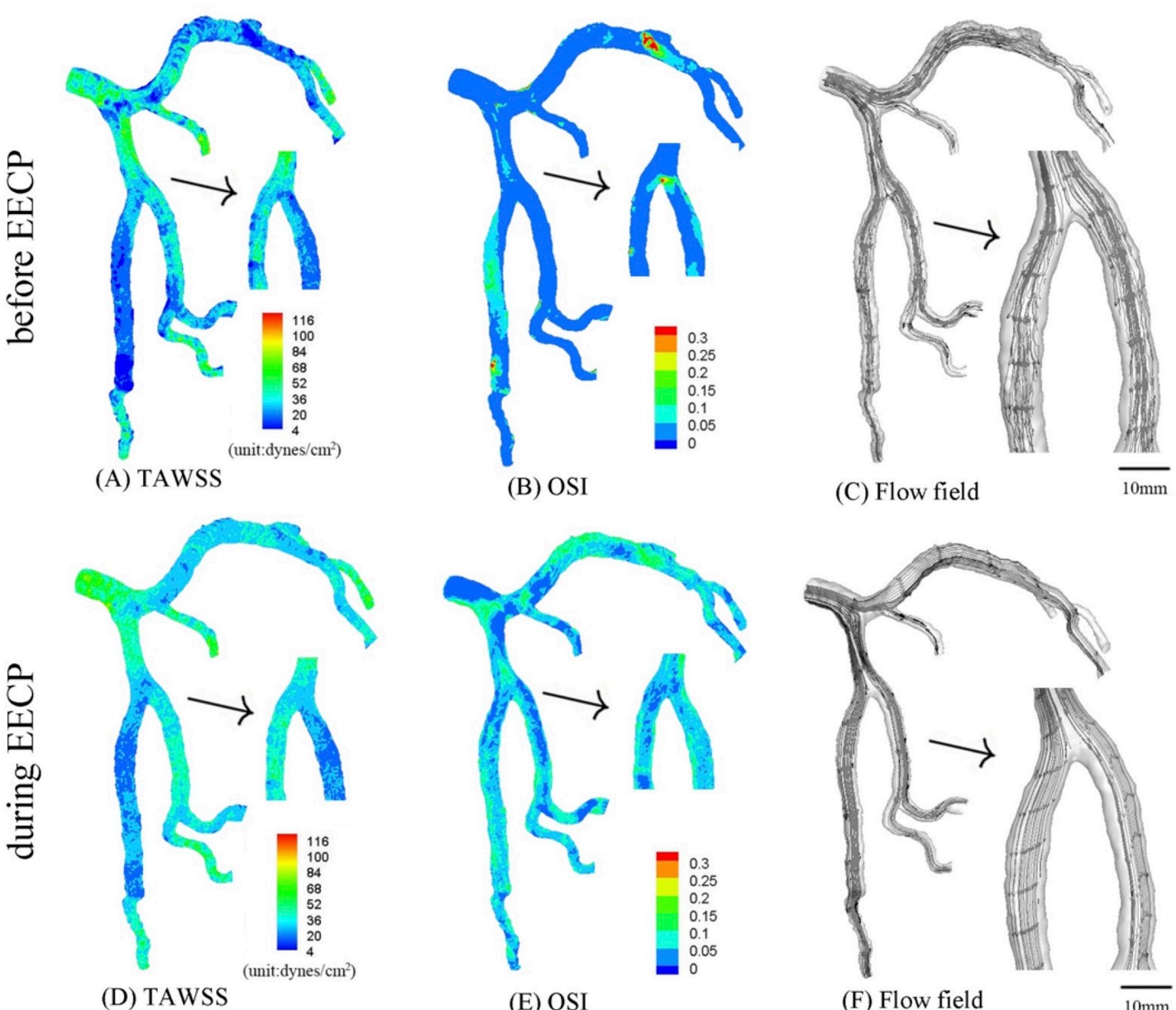

**Fig 4.** In correspondence with Fig 1A, TAWSS (A), OSI (B) and flow field (C) in the epicardial tree that has an idealized 50% area stenosis at the parent vessel (stenotic length of 7.3 mm) in the first bifurcation of LAD arterial tree before EECP; TAWSS (D), OSI (E) and flow field (F) in the same tree during EECP. The small figures for TAWSS and OSI show the posterior view. The small figures for flow field show the zoomed view.

equals to

$$\frac{\text{Surface area}_{OSI \geq 0.15}}{\text{Surface area near a bifurcation}} \times 100\% \tag{2}$$

, where surface area of OSI $\geq$ 0.15 is high-risk area) were computed at coronary bifurcations using the method in Ref. [31]. The average peak velocity along the main trunk of epicardial left anterior descending artery (LAD) was computed. The curve fitting in Fig 6 was presented using Matlab software (R2014a, MathWorks, USA).

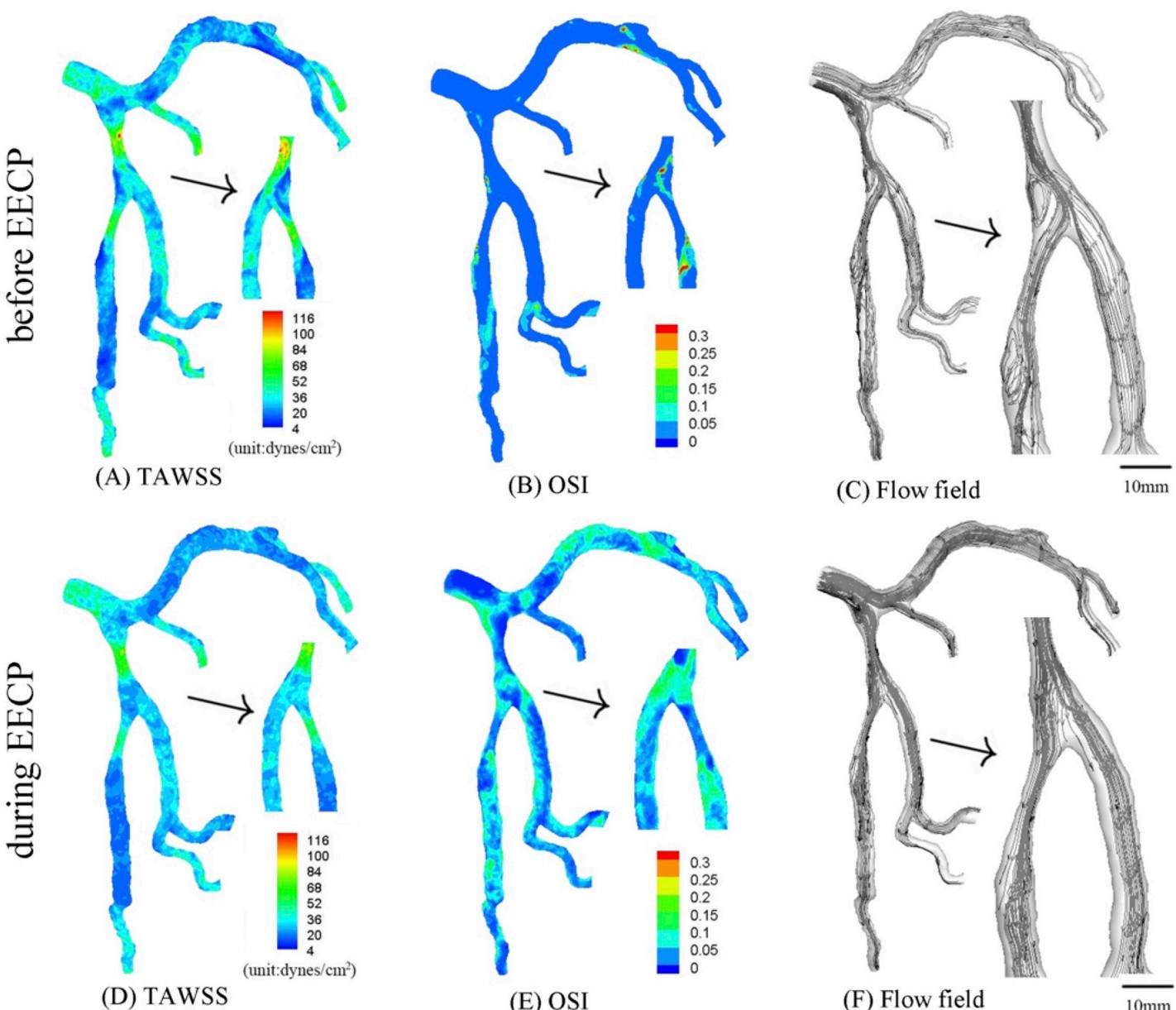

**Fig 5.** In correspondence with Fig 1A, TAWSS (A), OSI (B) and flow field (C) in the epicardial tree that has an idealized 75% area stenosis at the parent vessel (stenotic length of 7.3 mm) and an idealized 75% area stenosis at the large daughter vessel (stenotic length of 7.9 mm) in the first bifurcation of LAD arterial tree before EECP; TAWSS (D), OSI (E) and flow field (F) in the same tree during EECP. The small figures for TAWSS and OSI show the posterior view. The small figures for flow field show the zoomed view.

## Results

CFD simulations were performed in the epicardial LMCA trees of seven human subjects. A representative LMCA tree is shown in Fig 1A. Based on this tree, Figs 2–5 were constructed, with Figs 3–5 presenting several kinds of idealized stenoses. Fig 6 and Table 3 covered all seven patients who received EECP with different D/S values (0.26, 0.38, 0.65, 0.79, 0.92, 1.21, 1.72). All the trees in Fig 6 were original trees without idealized stenoses.

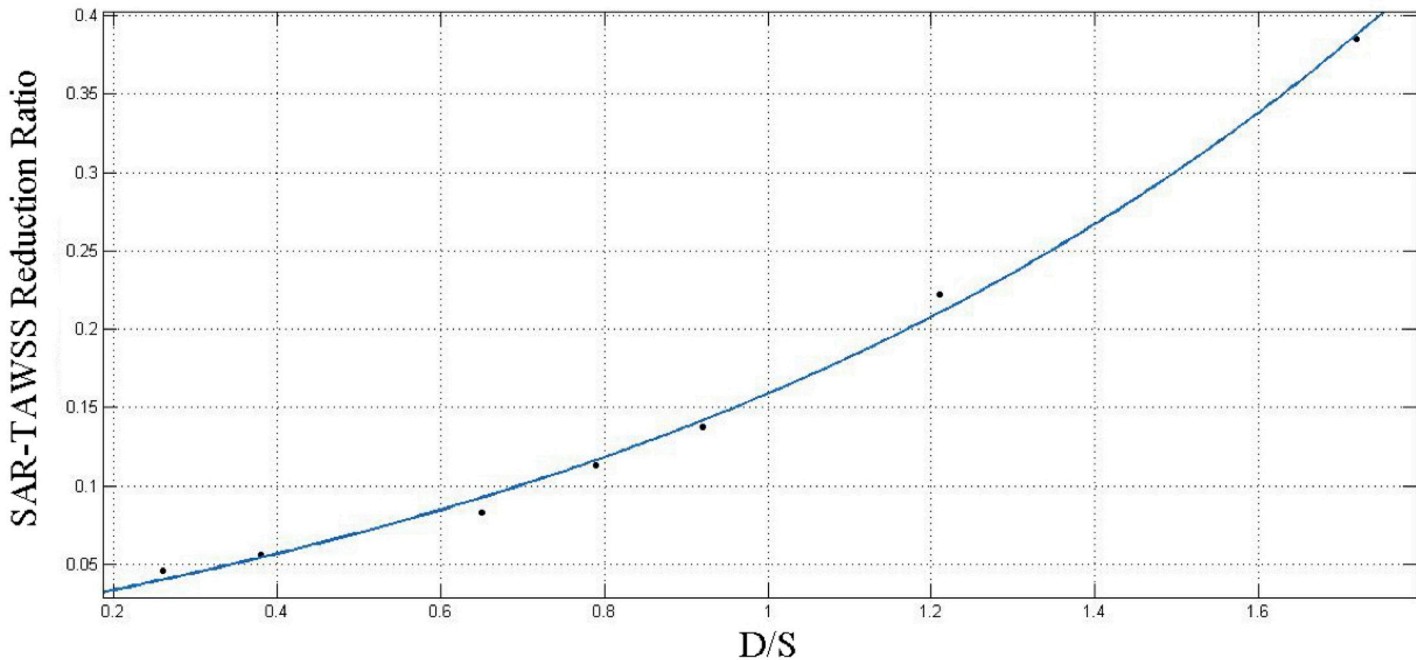

**Fig 6. The graph of fitted function of SAR-TAWSS reduction ratio (during EECP vs. before EECP) with D/S.**

The patient in Figs 1–5 received EECP with D/S of 1.21. The aortic pressure wave in Fig 2A was measured before the patient received EECP treatment, while the aortic pressure wave in Fig 2E was measured during EECP treatment. Fig 2B–2D show the distribution of TAWSS, OSI and flow field in the epicardial LMCA tree in Fig 1A. Fig 2F–2H show the distribution of those hemodynamic parameters in the epicardial LMCA tree under pressure in Fig 2E, which lead to decreased SAR-TAWSS (as shown in Table 1) and more regular flows downstream in the first bifurcation of LAD arterial tree (as shown in Fig 2H vs. Fig 2D). The corresponding average peak velocity along the epicardial LAD main trunk was presented in Table 2. In comparison with the case before EECP, the LAD main trunk decreased the pressure drop during EECP.

TAWSS and OSI were computed, and flow field was graphed in the epicardial LMCA tree that had an idealized severe stenosis (75% area stenosis) at the parent vessel in the first LAD bifurcation before and during EECP respectively, as shown in Fig 3A–3F. In comparison with original trees, the trees with severe stenosis demonstrated decreased SAR-TAWSS, increased SAR-OSI and complex flow patterns (significantly increased flow vortices and secondary flows distal to the stenosis). As shown in Fig 3D–3F and Tables 1 and 2, EECP at the tree with a severe stenosis significantly improved the hemodynamic conditions (i.e., significant decreased flow vortices and secondary flows distal to the stenosis) and reduced atherosclerosis-prone zones (i.e., about 50% reduction of SAR-TAWSS and SAR-OSI). Compared with severe stenosis, an idealized mild stenosis (50% area stenosis) was created at the parent vessel in the first LAD bifurcation, and the distribution of TAWSS, OSI and flow field (before EECP and during EECP, respectively) were shown in Fig 4A–4F.

Moreover, serial stenoses were created at the parent vessel (75% area stenosis, stenotic length of 7.3 mm) and large daughter vessel (75% area stenosis, stenotic length of 7.9 mm) in the first LAD bifurcation, and the distribution of TAWSS, OSI and flow field (before EECP and during EECP, respectively) were shown in Fig 5A–5F. Serial severe stenoses significantly

**Table 1. SAR-TAWSS and SAR-OSI at the mother vessel in the first bifurcation of LAD arterial tree.**

|  | SAR-TAWSS | SAR-OSI |
|---|---|---|
| Original tree before EECP | 7.2% | 0% |
| Original tree during EECP | 5.6% | 0% |
| 75% stenosis before EECP | 12.3% | 6.8% |
| 75% stenosis during EECP | 6.7% | 2.5% |
| 50% stenosis before EECP | 8.8% | 3.2% |
| 50% stenosis during EECP | 5.3% | 0% |
| Serial stenoses before EECP | 26.8% | 13.1% |
| Serial stenoses during EECP | 7.7% | 4.6% |

deteriorated hemodynamic conditions (i.e., increased flow vortices and secondary flows distal to both of the stenoses, as shown in Fig 5C) and significantly increased the peak pressure gradient along the epicardial LAD main trunk (as shown in Table 2). EECP at the tree with serial severe stenoses improved the hemodynamic condition to a more significant extent than the case of a single severe stenosis (Fig 5 vs. Fig 3, Tables 1 and 2).

Furthermore, the variation of SAR-TAWSS reduction ratio (during EECP vs. before EECP) in seven patients with different D/S (the real value of D/S during EECP they received) was shown in Table 3, with each row denoting a patient. When D/S was very small, the SAR-TAWSS reduction ratio was also very small, showing that small D/S could only improve the hemodynamic condition mildly. The SAR-TAWSS reduction ratio significantly increased as D/S became larger. The curve fitting of SAR-TAWSS reduction ratio (during EECP vs. before EECP) with D/S was presented in Fig 6. The exponential function $f(x) = ae^{bx}+c$ was used, where a = 0.0932, b = 0.9359 and c = -0.07876, with R-square value of 0.9966.

## Discussion

As shown in Fig 2B–2D, a normal tree with a mild stenosis (about 10% area stenosis) at the parent vessel in the first bifurcation of LAD artery led to mildly deteriorated hemodynamic conditions (i.e., decreased TAWSS and increased OSI) and complex flow patterns (increased flow vortices and secondary flows distal to the stenosis). In comparison with Fig 2B–2D, when

**Table 2. Average peak velocity along the epicardial LAD main trunk.**

|  | Average peak velocity (cm/s) | |
|---|---|---|
|  | Before EECP | During EECP |
| Original tree | Case 1[a] | Case 2[b] |
|  | 6.83 | 11.57 |
| 75% stenosis | Case 3[c] | Case 4[d] |
|  | 7.86 | 18.25 |
| 50% stenosis | Case 5[e] | Case 6[f] |
|  | 7.12 | 13.88 |
| Serial stenoses | Case 7[g] | Case 8[h] |
|  | 8.91 | 23.61 |

Case 1: original tree before EECP in Fig 2B; Case 2: original tree during EECP in Fig 2F

Case 3: a 75% stenosis before EECP in Fig 3A; Case 4: a 75% stenosis during EECP in Fig 3D

Case 5: a 50% stenosis before EECP in Fig 4A; Case 6: a 50% stenosis during EECP in Fig 4D

Case 7: serial stenoses before EECP in Fig 5A; Case 8: serial stenoses during EECP in Fig 5D

**Table 3. The variation of SAR-TAWSS reduction ratio (during EECP vs. before EECP) in seven patients with different D/S (the real value of D/S during EECP).**

| D/S | SAR-TAWSS reduction ratio |
|---|---|
| 0.26 | 4.6% |
| 0.38 | 5.6% |
| 0.65 | 8.3% |
| 0.79 | 11.3% |
| 0.92 | 13.8% |
| 1.21 | 22.2% |
| 1.72 | 38.5% |

the normal tree received EECP with D/S of 1.21 (Fig 2E), hemodynamic conditions and flow patterns could be improved, and mild secondary flows downstream of the first bifurcation was also observed (Fig 2H). The TAWSS near the first bifurcation of LAD artery (10mm length from the distal bifurcation to the large and the small daughter vessel) increased from 24.3 dynes/cm$^2$ (before EECP) to 46.7 dynes/cm$^2$ (during EECP), which was consistent with previous studies [1, 3, 27]. As shown in Table 1, SAR-TAWSS reduced from 7.2% to 5.6%, indicating that EECP may improve hemodynamic conditions and decrease atherosclerosis-prone zones (i.e., a decrease of SAR-TAWSS). Similar to previous studies [6, 27], the average peak velocity significantly increased from 6.83 cm/s to 11.57 cm/s during EECP. These results significantly demonstrate that EECP could improve endothelial function in coronary arteries by altering the hemodynamic conditions.

In comparison with the original trees, when the tree with a severe stenosis received EECP with D/S of 1.21, hemodynamic conditions and flow patterns could be restored to a more significant extent (Fig 3). Meanwhile, SAR-TAWSS reduced by 45.5% (the computational formula was (12.3%-6.7%)/(12.3%)) from 12.3% to 6.7%, and SAR-OSI reduced by 63.2% from 6.8% to 2.5%, as shown in Table 1. Therefore, EECP with large D/S may be beneficial for patients with severe atherosclerosis (i.e., by improving their hemodynamic conditions and flow patterns). However, EECP could only serve as an adjuvant therapy for severe atherosclerotic patients, because there were still strong secondary flows in the second LAD bifurcation (Fig 3F vs. Fig 2H), which may be a risk factor for restenosis after PCI and needs further investigation.

Compared with the case of severe stenosis, when the tree with a mild stenosis received EECP with D/S of 1.21 (Fig 2E), hemodynamic conditions and flow patterns were restored to a lesser extent. SAR-TAWSS reduced by 39.8% from 8.8% to 5.3%, and SAR-OSI reduced from 3.2% to 0% (but OSI at some area was very close to the critical value of 1.5). Meanwhile, for mild atherosclerotic patients, EECP could restore hemodynamic conditions with low values of SAR-TAWSS and SAR-OSI (i.e., SAR-OSI decreased to 0%, ref [36]) and more regular flow patterns (i.e., the decrease of flow vortices and secondary flows in Fig 4F vs. Fig 4C). Therefore, it was a relatively effective conservative treatment.

However, greater extent of improvement does not mean better therapeutic effect. For example, in the case of a severe stenosis, strong secondary flows still occurred (Fig 3F). In comparison, the improvement of hemodynamic conditions was more significant in original tree and mild stenosis (Fig 2H and Fig 4F), with no significant secondary flows or vortices and significant decrease of high-risk atherosclerotic area (SAR-TAWSS<6%, SAR-OSI = 0% during EECP). Therefore, the therapeutic effect of EECP could be better for a tree with a mild stenosis than a tree with a severe stenosis. Our findings are consistent with Chen et al., who found that "under this counterpulsation mode, the therapeutic effect became worse with the increased rate of coronary artery stenosis" [27].

Furthermore, the condition of the tree suffering serial severe stenoses was also analyzed. Serial severe stenoses significantly deteriorated hemodynamic conditions (i.e., increased flow vortices and secondary flows distal to both of the stenoses, as shown in Fig 5C), which may be a key risk factor for restenosis after PCI. When the tree was receiving EECP with D/S of 1.21 (Fig 2E), the hemodynamic condition was improved (i.e., the strong flow vortices and secondary flows almost disappeared, Fig 5F vs. Fig 5C), and atherosclerosis-prone zones reduced significantly (i.e., SAR-TAWSS reduced by 71.3% from 26.8% to 7.7%., and SAR-OSI reduced by 64.9% from 13.1% to 4.6%). Therefore, EECP with large D/S was also beneficial for severe atherosclerotic patients. Considering its effect in restoring the hemodynamic conditions of patients preparing for PCI, EECP was not only an excellent adjuvant therapy after PCI, but also might be an excellent conservative treatment before PCI.

D/S was an important parameter of EECP. In general, large value of D/S (>1.2) could significantly increase coronary perfusion pressure [6,7]. According to the reality of patients, the actual D/S was mostly lower than 1.2. However, the small D/S could still improve the hemodynamic conditions of patients (Table 3). Therefore, if a patient cannot receive EECP with a large value of D/S, which may make the patient uncomfortable or induce hypertension, he/she could choose EECP with a certain value of D/S suggested by the doctor based on the prediction data and experience. In this situation, EECP with small D/S would be an effective adjuvant therapy. As the value of D/S increases, the improvement of the hemodynamic conditions tends to be more significant (i.e., SAR-TAWSS reduction ratio increased rapidly, as shown in Table 3). More importantly, SAR-TAWSS reduction ratio even showed exponential growth with D/S (Fig 6). The study indicated that when a patient could sustain EECP with a large value of D/S (usually >1.2), the improvement in hemodynamic conditions could be very significant. However, more tests are needed to confirm this finding.

Several limitations need to be considered when interpreting the findings. In this study, the sample of patients was too small to find various degrees of coronary artery stenoses, so the idealized stenoses were created in LMCA trees. Different degrees of coronary artery stenoses should be divided into three groups (health trees, mild stenoses, and severe stenoses) in future studies. Besides, we used the aortic pressure waves to surrogate the inlet pressure waves of LMCA as the inlet boundary conditions. Because the pressure in the aorta is greater than that in the the LMCA, it may induce the overestimation of TAWSS. If the pressure waves of LMCA could be measured in the future, we can use more actual inlet boundary conditions. Moreover, the effects of non-newtonian fluid (especially the viscoelastic effect of blood) and elastic vessel walls were not considered in this study, which may lead to the overestimation of TAWSS [34]. More accurate models should be used in the future.

## Conclusions

A key finding of the study was that the improvement of hemodynamic conditions (i.e., average velocity, TAWSS, OSI and flow field) along the LMCA trees during EECP became more significant with the increase of D/S (approximately exponential growth) and the severity degree of stenosis at the bifurcations. Moreover, EECP with a low value of D/S (<1.2) could still improve the hemodynamic conditions of patients. The hemodynamic analysis in the epicardial coronary arterial tree improves our understandings of EECP as adjuvant therapy before or after PCI in patients with diffuse atherosclerosis.

## Supporting information

**S1 Appendix.** [37], [38], [39], [40], [41].
(DOCX)

**S1 File.**
(ZIP)

## Acknowledgments

Thanks to the anonymous colleagues in the Department of Radiology, Peking University Third Hospital for the collection and analysis of CT images.

## Author Contributions

**Conceptualization:** Ming Cui, Wei Gao, Dongguo Li, Wei Zhao.

**Data curation:** Ling Xu, Xi Chen, Chuan Ren, Haiyi Yu.

**Formal analysis:** Ling Xu, Xi Chen, Chuan Ren.

**Funding acquisition:** Xi Chen, Wei Zhao.

**Investigation:** Ling Xu, Xi Chen, Chuan Ren, Haiyi Yu, Dongguo Li, Wei Zhao.

**Methodology:** Ling Xu, Xi Chen, Dongguo Li, Wei Zhao.

**Project administration:** Ling Xu, Xi Chen, Dongguo Li, Wei Zhao.

**Resources:** Chuan Ren, Haiyi Yu, Wei Gao, Dongguo Li, Wei Zhao.

**Software:** Ling Xu, Xi Chen.

**Supervision:** Ming Cui, Wei Gao, Dongguo Li, Wei Zhao.

**Writing – original draft:** Ling Xu, Xi Chen.

**Writing – review & editing:** Dongguo Li, Wei Zhao.

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
