## [Decision Letter · Decision Letter 0]

27 Nov 2019

PONE-D-19-21646

The improvement of the shear stress of coronary arteries during Enhanced External Counterpulsation in patients with coronary heart disease

PLOS ONE

Dear Dr. Zhao,

Thank you for submitting your manuscript to PLOS ONE. After careful consideration, we feel that it has merit but does not fully meet PLOS ONE’s publication criteria as it currently stands. Therefore, we invite you to submit a revised version of the manuscript that addresses the points raised during the review process.

As you will see from the comments of the reviewer major points of critique were raised, especially regarding presentation of the manuscript and referencing of published literature.

We would appreciate receiving your revised manuscript within 2 months. To enhance the reproducibility of your results, we recommend that if applicable you deposit your laboratory protocols in protocols.io, where a protocol can be assigned its own identifier (DOI) such that it can be cited independently in the future. For instructions see: http://journals.plos.org/plosone/s/submission-guidelines#loc-laboratory-protocols

We look forward to receiving your revised manuscript.

Kind regards,

Rudolf Kirchmair

Academic Editor

PLOS ONE

Journal Requirements:

Additional Editor Comments (if provided):

Reviewers' comments:

Reviewer's Responses to Questions

**Comments to the Author**

1. Is the manuscript technically sound, and do the data support the conclusions?

Reviewer #1: Yes

2. Has the statistical analysis been performed appropriately and rigorously? 

Reviewer #1: Yes

3. Have the authors made all data underlying the findings in their manuscript fully available?

Reviewer #1: Yes

4. Is the manuscript presented in an intelligible fashion and written in standard English?

Reviewer #1: No

5. Review Comments to the Author

Reviewer #1: The present study is a numerical simulation on intracoronary hemodynamics caused by enhanced external counterpulsation (EECP). Authors used clinically measured aortic pressure and flow resistance model as the boundary conditions for the calculation of CFD. Through the analysis of various model with differing boundary (the ratio of diastolic pressure and systolic pressure (D/S)), some quantitative results and conclusions were acquired. It is an interesting work. However, some questions must be answered before the consideration for publishing in the PLOS ONE.

Major comments:

1. The present study calculated the variation of both wall shear stress (WSS) and oscillatory shear index (OSI) Shear stress before and during EECP. Why only WSS was emphasized in the title but not OSI?

2. In Abstract: oscillatory shear index (OSI) was mentioned in the Methods section, while no data were presented to demonstrate the variation of OSI before and during EECP and its impact on the coronary heart disease.

3. In Methods, it is necessary to describe that how the aortic pressure was measured.

4. In Methods, more details should be presented to explain the outlet boundary condition (flow resistances model).

5. Authors declared that the ratio of diastolic pressure and systolic pressure (D/S) is an important acute indicator during EECP. A reference will be helpful here (see doi: 10.1007/s11517-018-1834-z).

6. In Discussion, authors declared that the finding in this paper was consistent with the previous study (Chen et al.). However, there seems no conflict between your conclusions. Please check it carefully.

Minor comments:

1. In line 31, page 3, the comma before “The aortic pressure wave…” should be full stop.

2. In line 40, page 3, the reference [4] should be at the end of the sentence.

3. In line 43, page 3, the abbreviation “CHD” should be explained when it appeared first time in the main text.

4. In line 61, page 4, the sentence should be revised to “Recently, the CFD methods have been adopted to non-invasively determine the fractional flow reserve (FFR)”.

5. The references should be numbered so as to be reviewed conveniently.

6. There are other grammar, spelling and format mistakes, please revised the manuscript carefully.

6. PLOS authors have the option to publish the peer review history of their article (what does this mean?). If published, this will include your full peer review and any attached files.

Reviewer #1: No

---

## [Author Response · Author response to Decision Letter 0]

28 Dec 2019

Thanks to the professional comments given by the reviewer, we have made corresponding changes to the manuscript and benefited a lot from re-examination.

---

## [Decision Letter · Decision Letter 1]

20 Jan 2020

PONE-D-19-21646R1

The improvement of the shear stress and oscillatory shear index of coronary arteries during Enhanced External Counterpulsation in patients with coronary heart disease

PLOS ONE

Dear Dr. Zhao,

Thank you for submitting your manuscript to PLOS ONE. After careful consideration, we feel that it has merit but does not fully meet PLOS ONE’s publication criteria as it currently stands. Therefore, we invite you to submit a revised version of the manuscript that addresses the points raised during the review process.

As you will recognize from the comments of the reviewer some remaining minor points of critique were raised, especially regarding presentation of data.

We would appreciate receiving your revised manuscript within 2 months. To enhance the reproducibility of your results, we recommend that if applicable you deposit your laboratory protocols in protocols.io, where a protocol can be assigned its own identifier (DOI) such that it can be cited independently in the future. For instructions see: http://journals.plos.org/plosone/s/submission-guidelines#loc-laboratory-protocols

We look forward to receiving your revised manuscript.

Kind regards,

Rudolf Kirchmair

Academic Editor

PLOS ONE

Reviewers' comments:

Reviewer's Responses to Questions

**Comments to the Author**

1. If the authors have adequately addressed your comments raised in a previous round of review and you feel that this manuscript is now acceptable for publication, you may indicate that here to bypass the “Comments to the Author” section, enter your conflict of interest statement in the “Confidential to Editor” section, and submit your "Accept" recommendation.

Reviewer #1: All comments have been addressed

2. Is the manuscript technically sound, and do the data support the conclusions?

Reviewer #1: Yes

3. Has the statistical analysis been performed appropriately and rigorously? 

Reviewer #1: Yes

4. Have the authors made all data underlying the findings in their manuscript fully available?

Reviewer #1: No

5. Is the manuscript presented in an intelligible fashion and written in standard English?

Reviewer #1: Yes

6. Review Comments to the Author

Reviewer #1: After revision, the manuscript was significantly improved. However, there is still a minor question about the paper. It is acceptable for Plos One after the further revision.

Minor comments:

In page 6 of Methods, authors introduced the flow resistance model in the outlet boundary. However, no data was presented to be examined. The value of resistances are supposed to be available.

Besides, a suggestion for authors is that they should answer the questions point to point in a separate document for reviewing in their future submission.

7. PLOS authors have the option to publish the peer review history of their article (what does this mean?). If published, this will include your full peer review and any attached files.

Reviewer #1: No

---

## [Author Response · Author response to Decision Letter 1]

23 Jan 2020

Response: We revised the Appendix to elaborate on the resistance boundary condition. It is very difficult for current experimental techniques to non-invasively and accurately determine the boundary conditions at the outlet of patient epicardial coronary arterial trees. Many computational approaches (e.g., growth of distal subtrees, multiple-element Windkessel model, hybrid one-dimensional/Womersley model, scaling laws, and so on) have been proposed to determine the outlet boundary conditions. There is debate on these methods, however, in comparison with invasive measurements. Gijsen’s group has recently shown the advantage of the scaling law approach (J. Biomech. 44:1089, 2011). Hence, we have selected the scaling law to determine the value of resistances as the outlet boundary conditions. 

In addition, we are very grateful for the reviewer's comments, and we must pay attention to answering the questions point to point in separate files in the future.

---

## [Decision Letter · Decision Letter 2]

24 Feb 2020

The improvement of the shear stress and oscillatory shear index of coronary arteries during Enhanced External Counterpulsation in patients with coronary heart disease

PONE-D-19-21646R2

Dear Dr. Zhao,

We are pleased to inform you that your manuscript has been judged scientifically suitable for publication and will be formally accepted for publication once it complies with all outstanding technical requirements.

With kind regards,

Rudolf Kirchmair

Academic Editor

PLOS ONE

Additional Editor Comments (optional):

Reviewers' comments:

Reviewer's Responses to Questions

**Comments to the Author**

1. If the authors have adequately addressed your comments raised in a previous round of review and you feel that this manuscript is now acceptable for publication, you may indicate that here to bypass the “Comments to the Author” section, enter your conflict of interest statement in the “Confidential to Editor” section, and submit your "Accept" recommendation.

Reviewer #1: All comments have been addressed

2. Is the manuscript technically sound, and do the data support the conclusions?

Reviewer #1: Yes

3. Has the statistical analysis been performed appropriately and rigorously? 

Reviewer #1: Yes

4. Have the authors made all data underlying the findings in their manuscript fully available?

Reviewer #1: Yes

5. Is the manuscript presented in an intelligible fashion and written in standard English?

Reviewer #1: Yes

6. Review Comments to the Author

Reviewer #1: (No Response)

7. PLOS authors have the option to publish the peer review history of their article (what does this mean?). If published, this will include your full peer review and any attached files.

Reviewer #1: No

---

## [Editor Report · Acceptance letter]

9 Mar 2020

PONE-D-19-21646R2 

The improvement of the shear stress and oscillatory shear index of coronary arteries during Enhanced External Counterpulsation in patients with coronary heart disease 

Dear Dr. Zhao:

I am pleased to inform you that your manuscript has been deemed suitable for publication in PLOS ONE. Congratulations! Your manuscript is now with our production department. 

With kind regards,

on behalf of

Prof Rudolf Kirchmair 

Academic Editor

PLOS ONE